# Dental age prediction from panoramic radiographs using machine learning techniques

**Mehdi Salehizeinabadi**[1]*, **Nazila Ameli**[1], **Kasra Kouchehbaghi**[2], **Sara Arastoo**[3], **Saghar Neghab**[4], **Ida M. Kornerup**[1], **Camila Pacheco-Pereira**[1]

**1** Mike Petryk School of Dentistry, University of Alberta, Edmonton, Alberta, Canada, **2** Private Practice, Toronto, Ontario, Canada, **3** Private Practice, Calgary, Alberta, Canada, **4** School of Dentistry, University of Toronto, Ontario, Canada

* zeianabad@ualberta.ca

## Abstract

Dental age (DA) estimation is a key diagnostic tool in pediatric dentistry, particularly when birth records are unavailable or unreliable. It guides decisions on growth assessment, orthodontic planning, and timing of interventions such as space maintenance or extractions. Unlike skeletal maturity, dental development is less affected by nutritional and environmental factors, making it a reliable marker of biological age. Conventional methods require expert interpretation and are prone to variability. There is growing interest in automated, objective approaches to streamline this process and enhance clinical utility. A total of 550 panoramic radiographs from children aged 3–14 years were labeled into 11 dental age groups based on the AAPD reference chart by two experienced pediatric dentists. Images with poor quality were excluded. The dataset was divided into training (80%) and validation (20%) sets, with data augmentation applied to the training set. The YOLOv11n-cls model, consisting of 86 layers and 1.54 million parameters, was trained for 30 epochs using the Ultralytics engine and AdamW optimizer. Model performance was evaluated using Top-1 and Top-5 accuracy on the validation set and tested on an independent set of 203 images. Grad-CAM was used for model interpretability. The model achieved 92.6% Top-1 and 99.5% Top-5 accuracy on the validation set. Performance on the test set remained high, with most misclassifications occurring between adjacent age groups. Grad-CAM visualizations showed attention to clinically relevant areas like erupting molars and root development. The findings support the high performance of DL, through YOLOv11 for pediatric age prediction. The AI tool enabled fast, accurate, and interpretable DA classification, making it a strong candidate for clinical integration as an adjunct tool into pediatric dental practice.

**Data availability statement:** The data that support the findings of this study are available from the University of Alberta but restrictions apply to the availability of these data, which were used under license for the current study, and so are not publicly available. Data are however available at dentrsch@ualberta.ca upon reasonable request and with permission of University of Alberta.

**Funding:** The author(s) received no specific funding for this work.

**Competing interests:** The authors have declared that no competing interests exist.

## Author summary

Understanding a child's dental age is crucial for making informed decisions about treatments like braces, extractions, or space maintainers. Traditionally, estimating dental age requires dentists to manually examine X-rays and compare them to reference charts. This process can be time-consuming and subjective, especially when done by less experienced clinicians. In our study, we developed and tested an artificial intelligence (AI) model using panoramic dental X-rays of children aged 3–14. The model was trained to automatically estimate dental age based on visual features in the X-rays, like how a pediatric dentist would assess development. We used a cutting-edge algorithm called YOLOv11, which is known for its speed and accuracy. Our model was highly accurate and could also highlight the parts of the image it used to make predictions, helping build trust in its decisions. We believe this technology can assist dentists, especially in busy or underserved clinics, by providing quick and reliable age estimates that support better treatment planning for children.

## Introduction

Dental age (DA) estimation plays a central role in pediatric and forensic dentistry, especially when birth records are missing or unreliable [1,2]. It is widely used in clinical, legal, and anthropological contexts. Among biological age markers, dental development is considered highly reliable due to its low sensitivity to malnutrition and environmental factors [3–5]. Tooth formation and eruption follow a predictable pattern from early in utero to the third decade of life, supporting consistent age estimation [6,7].

Demirjian's method [8–10], based on staging mandibular teeth in panoramic radiographs, remains widely used, though population differences limit its universal applicability. Other traditional approaches, such as those by Nolla [11], Cameriere [12], and Willems [13], also depend on manual staging or linear measurements. These methods require expert input and often rely on regression models, limiting scalability and introducing observer variability. Moreover, conventional approaches show reduced reliability during mixed dentition phases, which can affect accuracy and clinical sensitivity in pediatric populations.

With recent advances in artificial intelligence (AI), particularly deep learning (DL), dental image analysis has become more automated and accurate [14–16]. Convolutional neural networks (CNNs) enable rapid and reproducible interpretation of dental radiographs [17–21]. Panoramic radiographs are ideal for these models, offering full dentition views with minimal radiation [22,23]. CNNs can detect dental maturation features such as crown/root formation and tooth germs [24], reducing subjectivity in DA estimation [25].

A recently published meta-analysis confirmed the high accuracy and clinical value of AI in radiographic DA estimation, further supporting DL adoption in dental practice

[26]. CNNs have also been successful in segmentation, classification, and detecting dental conditions such as caries and bone loss [27–32]. Despite progress, most AI models for DA estimation rely on complex, multi-stage pipelines [33] or focus on predicting chronological rather than dental age [22]. Even when machine learning enhances traditional methods like Demirjian or Cameriere, they still require manual inputs and regression outputs, limiting automation [7].

To overcome the present gaps in the literature, we propose using You Only Look Once version 11 (YOLOv11), a single-stage object detection algorithm optimized for speed and accuracy in medical imaging [34]. YOLOv11 has shown excellent performance in dental applications, achieving up to 97% precision and strong F1-scores in predicting third molar extraction difficulty [34] and detecting caries in pediatric patients [35], with low inference times [36]. No prior study has applied YOLOv11 for classifying DA in children aged 3–14, a crucial developmental period. This study will develop and validate a YOLOv11-based model using expert-labeled panoramic radiographs aligned with AAPD standards. We hypothesize that this model will outperform traditional and multi-stage approaches in accuracy, efficiency, and scalability.

## Materials and methods

### Ethics statement

This study was reviewed and approved by the University of Alberta Research Ethics Board (REB) under the ethics ID Pro00143202. The panoramic radiographs were obtained from the Oral and Maxillofacial Radiology service and were de-identified prior to analysis. Because the dataset consisted of historical clinical radiographs without any personal health identifiers, and no contact with patients was required, the University of Alberta REB waived the requirement for informed consent.

### Dataset augmentation and preprocessing

The dataset consisted of 550 panoramic radiographs from children aged 3–14 years, categorized into 11 DA groups based on expert labelling aligned with the AAPD reference chart. The frequency of each group is presented in Table 1.

To ensure consistency and quality, radiographs with significant artifacts, low resolution, or overlapping anatomical structures were excluded. These cases were considered potential outliers, as their inclusion could disproportionately influence model learning, and were removed during preprocessing. Images were manually labelled by two pediatric dentistry experts, each with over ten years of experience. In cases of disagreement, a third pediatric dentist reviewed the radiographs and provided the final ground truth labels.

Table 1. Distribution of panoramic radiographs across the 11 age groups.2.

| Age Group (Years) | Number of Images (%) |
| --- | --- |
| 3–4 | 17 (3.1) |
| 4–5 | 15 (2.7) |
| 5–6 | 19 (3.5) |
| 6–7 | 28 (5.1) |
| 7–8 | 41 (7.5) |
| 8–9 | 66 (12.1) |
| 9–10 | 83 (15.2) |
| 10–11 | 75 (13.7) |
| 11–12 | 71 (13.0) |
| 12–13 | 66 (12.1) |
| 13–14 | 69 (12.6) |
| Total | 550 (100) |

To prevent data leakage, the dataset was first divided into training (80%, 440 images) and validation (20%, 110 images) sets before any augmentation was applied. This approach ensured that augmented variants of the same image were not distributed across different subsets. To enhance generalizability and reduce the risk of overfitting, the training set subsequently underwent three rounds of data augmentation. Techniques included horizontal flipping (probability = 0.5), random rotations (±15°), translations (up to 10% of image dimensions), brightness variation (±20%), contrast variation (±20%), and random erasing with a probability of 0.4 (erasure size up to 20% of image area). All images were resized to 320 × 320 pixels to conform to the model's input specifications. Following augmentation, the training set was expanded to include 1,320 images.

## Model architecture

We employed the YOLOv11n-cls model [34], a lightweight, real-time, single-stage DL architecture tailored for classification tasks. The model comprises 86 layers and approximately 1.54 million parameters, designed to balance high accuracy with computational efficiency. Internally, the architecture integrates C3k2 convolutional blocks, which facilitate deep feature extraction while maintaining a compact design. Additionally, the model incorporates a Cross-Stage Partial Spatial Attention (C2PSA) module, which enhances attention to salient regions of the image by distributing focus across spatial hierarchies. The final classification head outputs probabilities for each of the 11 DA categories, allowing the model to assign a label to each input image based on learned patterns of dental development.

## Training procedure

Model training was conducted using the Ultralytics v8.3.137 engine on a Tesla T4 GPU, utilizing CUDA acceleration and Automatic Mixed Precision (AMP) to enhance computational efficiency. The AdamW optimizer was automatically selected during training initialization, with a starting learning rate of 0.000667 and momentum set to 0.9. The training process spanned 30 epochs with a batch size of 16 and included real-time logging of performance metrics such as classification loss and accuracy. Model checkpoints were saved periodically, and the best-performing version was used for final evaluation. To address class imbalance across age groups, class weights were applied within the loss function, ensuring that underrepresented categories contributed proportionally more to the optimization process. This adjustment reduced the bias toward majority classes and complemented our targeted augmentation strategies.

## Performance evaluation

Model performance was evaluated using Top-1 and Top-5 accuracy metrics on the validation dataset. Top-1 accuracy reflects the proportion of instances where the model's highest-confidence prediction matched the ground truth, while Top-5 accuracy measures whether the correct label appeared among the top five predicted probabilities.

## Explainability via Grad-CAM

To gain insights into the model's decision-making process and assess the clinical relevance of its focus areas, we applied Gradient-weighted Class Activation Mapping (Grad-CAM), [37] Using the pytorch-grad-cam library, we visualized the regions in each radiograph that most influenced the model's classification decision. The final convolutional layer (model [10].conv) was selected as the target layer for visualization. After predicting the class for a given radiograph, Grad-CAM was applied to generate a heatmap that highlighted areas of strong model activation. This heatmap was then overlaid on the original radiograph to produce an interpretable visualization of model attention. The analysis confirmed that the model frequently focused on biologically meaningful regions such as the roots, crown formation, and eruption zones of premolars and molars, areas clinically relevant to dental age assessment.

## Statistical analysis

To evaluate inter-rater reliability between two expert annotators who labeled the age groups, Cohen's kappa and weighted Cohen's kappa were calculated. These metrics quantify the degree of agreement beyond chance and are particularly appropriate for categorical and ordinal data [38]. Performance metrics were recorded at each epoch to monitor convergence and generalization. Results were obtained using the best model checkpoint, selected based on validation accuracy. In addition to the validation set, an independent test set consisting of 203 new panoramic radiographs, unseen during training, was used to further assess the model's generalizability and real-world applicability.

Weighted kappa was used to account for the ordinal nature of age group categories, applying less penalty to near-disagreements than to distant ones [39]. Model performance was assessed using Top-1 and Top-5 accuracy, which indicate the proportion of test samples where the correct class was ranked first or within the top five predictions, respectively. These metrics are widely used in multi-class classification tasks to evaluate model precision under both strict and relaxed criteria.

In accordance with reporting standards for AI-based prediction models, the completed TRIPOD-AI checklist [40] has been provided as Supplementary Material (S1 File).

## Results

A total of 753 unique panoramic radiographs were included in this study: 550 images used for training/validation and 203 images reserved for independent testing. Of the 550 images, 440 were allocated to the training set (which was subsequently expanded to 1,320 images through augmentation) and 110 were retained as the validation set. Inter-rater agreement between the two evaluators was high, with Cohen's kappa = 0.921 and weighted Cohen's kappa = 0.967, indicating excellent consistency in age group labeling across the dataset.

Model training progressed smoothly over 30 epochs, with both training and validation loss decreasing consistently from initial values above 2.0 to final values below 0.3. Early stabilization in the loss curve was observed around epoch 10, with performance remaining stable through the remainder of training (Fig 1). The consistent convergence of training and validation loss further indicates stable learning dynamics and minimal overfitting.

On the validation dataset, the model achieved a final Top-1 accuracy of 92.6%, indicating that the predicted age group exactly matched the ground truth in nine out of ten cases. Additionally, the Top-5 accuracy reached 99.5%, demonstrating that the correct age group was consistently among the top five predictions. These findings reflect the model's strong discriminative capability across a relatively fine-grained classification task involving 11 discrete age groups.

The confusion matrix in Fig 2 provides a detailed breakdown of the model's predictions across all dental age groups. While overall accuracy was high, most misclassifications were concentrated between adjacent age categories (e.g., 7–8 vs. 8–9 years, and 11–12 vs. 12–13 years), reflecting the subtle developmental changes that occur in close chronological intervals.

Grad-CAM visualizations further confirmed the model's clinical validity. In representative samples, the generated activation maps overlapped with key dental structures involved in chronological development. These included areas where root apex formation, crown completion, and dental eruption patterns were visible, features commonly used by pediatric dentists to estimate dental age. One example visualization is presented in Figs 3–5, illustrating the model's attention distribution for a correctly classified radiograph in the 4–5, 7–8, and 11–12-year age groups, respectively. The model's focus aligned with regions of active premolar and second molar development, providing qualitative validation for the model's predictions.

## Discussion

Knowing a child's DA helps guide interventions such as interceptive orthodontics, space maintainers, and extractions, ultimately improving long-term oral health outcomes [41]. This study demonstrates that YOLOv11 can accurately predict dental age estimation in children aged 3–14 years based on panoramic radiographs. Its fully automated nature and high

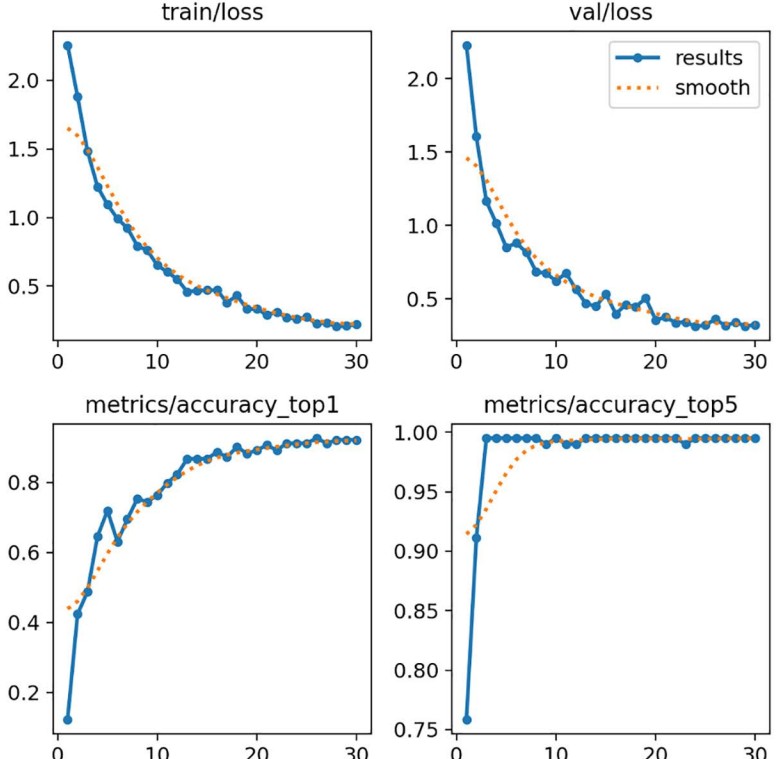

**Fig 1. Training progresses over 30 epochs.** (Top left) Training loss, (Top right) Validation loss, (Bottom left) Top-1 accuracy, (Bottom right) Top-5 accuracy. Solid lines represent recorded values, and dotted lines indicate smoothed trends.

performance, even with a modest dataset make it a strong candidate for clinical integration. The model achieved a high overall accuracy (92.6%), demonstrating strong potential for real-world clinical application in pediatric dentistry.

YOLOv11 has demonstrated superior results in dental applications. For example, it achieved 97% precision and 95.76% F1-score in assessing mandibular third molar extraction difficulty from panoramic X-rays [34], and outperformed previous YOLO versions in detecting caries and restorations in pediatric images [35]. Compared to two-stage detectors like Faster R-CNN and Mask R-CNN, YOLOv11 balances speed and accuracy, with an inference time as low as 13.5 ms [36].

A key strength of this study lies in the fully automated design of the model. Unlike traditional models, DL techniques like YOLOv11 offer objective, efficient, and scalable solutions that remove the burden of manual analysis [42].

Several recent studies have applied DL models to dental age estimation. For instance, Lee et al. trained CNNs using manually annotated developmental stages based on Demirjian's criteria [43] while Roh et al. used neural networks to generate regression equations from tooth stages on the mandible [44]. However, most of these models still relied on manual annotations, regression-based predictions, or multi-stage architectures.

To the best of our knowledge, this is the first study to employ YOLOv11 for direct classification of pediatric panoramic images into dental age groups without the need for image annotation or region of interest segmentation, offering a streamlined and fast alternative to prior approaches. Although we used only a single network architecture, the model achieved comparable or better accuracy than studies with more complex designs. For example, Roh et al. reported mean average error (MAE)s around 0.5–0.6 using manually labeled stages and regression models [44], whereas our model achieved 92% classification accuracy using a purely image-based approach with minimal preprocessing.

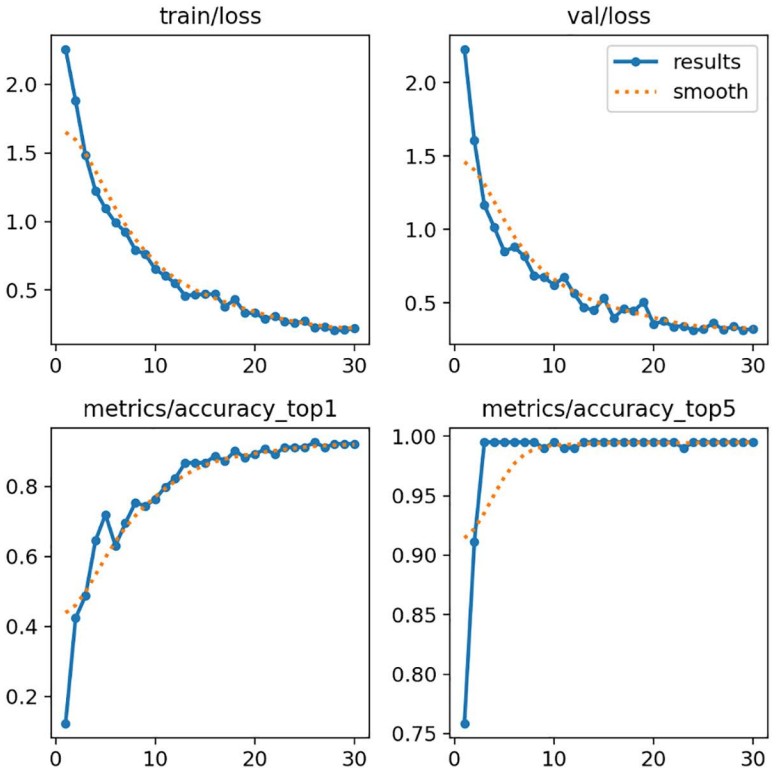

**Fig 2. Confusion matrix illustrating the model's performance across 11 dental age categories on a test set of 203 panoramic radiographs.**

Unlike traditional methods that rely on annotated biological markers [7,44], our model learned directly from expert-generated labels, enabling it to autonomously extract clinically relevant features from panoramic radiographs. This approach reduces chairside time, minimizes human error, and is well-suited for routine use, particularly in low-resource settings. The data-driven strategy allowed the model to uncover latent developmental patterns that contributed to its high accuracy. Grad-CAM visualizations confirmed the model's focus on meaningful anatomical features such as root resorption, calcification, and maturation without explicit guidance, supporting clinician confidence. This feature not only fosters trust but also enhances education and shared decision-making with patients and caregivers. Closer examination showed that these errors were typically associated with borderline anatomical features such as overlapping stages of molar eruption and root development, which are difficult to distinguish even for expert clinicians. This suggests that the misclassifications reflect biological variability more than algorithmic shortcomings. To mitigate such errors, future work could benefit from larger datasets with greater representation of transitional stages, adoption of ordinal classification/regression frameworks, or incorporation of multimodal data (e.g., combining radiographs with clinical records). Clinically, these misclassifications are not significant because treatment planning decisions rarely depend on differences of less than one year in dental age. This supports the robustness and practical utility of our proposed approach despite minor variability at group boundaries.

In contrast to previous studies such as Roh et al. [44], which limited their model input to the mandibular dentition and reported an MAE of 0.501, our approach integrates the entire panoramic image including both maxillary and mandibular arches, this broader context enables improved accuracy in borderline cases and supports comprehensive treatment planning. Clinically, such precision can enhance decision-making around critical interventions like interceptive orthodontics, space maintenance, and growth assessment, all of which hinge on accurate dental age rather than chronological age.

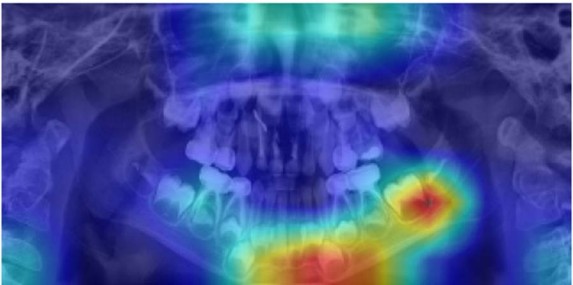

**Fig 3. Grad-CAM visualization of a panoramic radiograph classified into dental age group 4-5.** The heatmap highlights regions of high model attention, particularly around the developing molars and lower incisors, indicating the anatomical features that most influenced the model's classification decision.

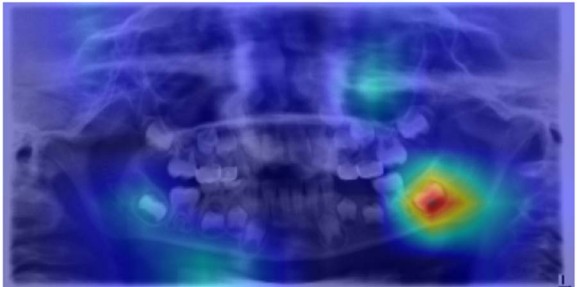

**Fig 4. Grad-CAM visualization of a panoramic radiograph classified into dental age group 7-8.** The heatmap highlights regions of high model attention, particularly around the developing second molars, indicating the anatomical features that most influenced the model's classification decision.

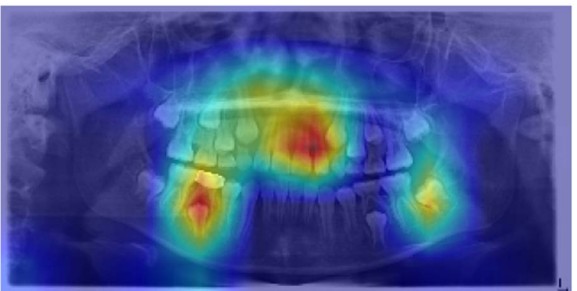

**Fig 5. Grad-CAM visualization of a panoramic radiograph classified into dental age group 11-12.** The heatmap highlights regions of high model attention, particularly around the developing second molars, premolars, and canines indicating the anatomical features that most influenced the model's classification decision.

Unlike the quadrant-based approach of Kurniawan et al. [45], which required manual cropping and reported 74% accuracy through classification, our model achieved higher precision through automated full-image classification without the need for manual cropping or regression-based pipelines.

From a technical standpoint, we focused on optimizing the YOLOv11 architecture without comparing alternative models due to computational limitations. However, YOLOv11 was selected based on its state-of-the-art performance in prior

dental imaging studies, and our model outperformed or matched the accuracy of more complex multi-stage or regression-based methods in the literature. The choice of 11 finely stratified age groups also reflects clinical relevance and enhances interpretability for pediatric dental planning.

We did not include comparisons with conventional machine learning methods (e.g., SVM, k-NN, random forests) because these approaches typically require handcrafted feature extraction, which introduces observer bias and limits performance in high-dimensional image data. In contrast, deep learning models such as YOLOv11 can automatically learn clinically relevant features directly from raw panoramic radiographs, an advantage highlighted in recent studies [17,19,44]. Moreover, traditional ML approaches have shown limited scalability in dental imaging tasks despite feature engineering, whereas DL models maintain accuracy with raw input. While a direct comparison would be scientifically valuable, it was beyond the scope of the present study, which focused on evaluating the feasibility of a fully automated DL pipeline. We propose this as a direction for future research.

This study presents some limitations primarily related to dataset composition. First, although the dataset size (550 images) may appear modest, it was carefully curated from clinically justified cases, and each image was labeled by two experienced pediatric dentists with adjudication for any discrepancies. This ensured high-quality ground truth, which has been shown to be more critical than sheer volume in early-phase AI development. Moreover, performance was validated on a completely independent test set of 203 unseen images, demonstrating strong generalizability and reinforcing the robustness of our findings despite dataset size constraints. In addition, the dataset exhibited imbalance across age groups, which may have influenced predictive performance for categories with fewer samples. This imbalance reflects real-world clinical patterns, as OPGs are less frequently indicated in younger children due to clinical guidelines and practical considerations. Although our model maintained high overall accuracy despite these disparities, future studies should validate performance on larger, more balanced, multi-center datasets to confirm generalizability across all age groups. To mitigate the impact of class imbalance, we applied targeted augmentation to minority age groups (e.g., rotations, translations, brightness/contrast adjustments) and incorporated class weights into the loss function, allowing underrepresented categories to contribute more strongly to the optimization process. These strategies improved generalization across unevenly distributed classes. Nevertheless, future work should explore additional approaches, such as focal loss, oversampling, or ensemble methods, to further enhance performance in highly imbalanced datasets.

Second, the radiographs were obtained from a single academic center using standardized imaging acquisition protocols. While this limits geographic and equipment variability, it also reduces confounding from imaging heterogeneity, allowing clearer interpretation of model performance. The consistently high Top-1 and Top-5 accuracy across internal and external test sets suggests that the model learned clinically generalizable features rather than overfitting to specific image acquisition characteristics. However, before clinical implementation, external validation on more diverse images centers/institutions remains a future goal. However, the AAPD-based labeling framework used here aligns with internationally accepted dental development standards, increasing the potential for broader applicability of the model.

Another point to note is the potential influence of outlier cases. In this study, outliers were primarily radiographs of insufficient quality or with significant artifacts, which were excluded during preprocessing. While this ensured a consistent dataset, future work could explore automated quality-control pipelines to systematically identify and manage such outliers in larger multi-institutional datasets.

Although this study focused exclusively on YOLOv11, we acknowledge that comparative evaluation with other DL architectures, such as ResNet, EfficientNet, or DenseNet, would provide additional insights into relative performance. Our choice of YOLOv11 was based on its recent architectural improvements, including C2PSA and C3k2 modules, which enhance feature learning and efficiency in medical imaging tasks. Furthermore, YOLOv11 has demonstrated strong results in dental applications, including caries detection and surgical risk assessment [35,36]. Future work should directly benchmark YOLOv11 against other CNN-based models under standardized preprocessing and evaluation pipelines to validate its comparative advantages.

Another limitation of this study is the absence of a formal calibration analysis. While the model demonstrated high classification accuracy, calibration metrics such as reliability diagrams, Brier scores, or expected calibration error were not evaluated. Future studies should incorporate these measures to assess how well the predicted probabilities align with true outcomes, thereby strengthening confidence in the model's clinical applicability.

Lastly, although explainability was limited to Grad-CAM visualizations, these consistently aligned with key anatomical regions used in manual age estimation, supporting the clinical validity of model attention. Future work will aim to expand on these insights using quantitative explainability frameworks and multi-institutional data to further reinforce model trustworthiness.

In summary, while certain limitations exist, they were addressed through careful study design, expert labeling, and robust internal and external testing. The results underscore the model's strong performance, clinical relevance, and potential for scalable application in pediatric dental settings.

The potential for clinical translation should also be discussed. Given that many pediatric dental treatments including space maintenance, pulp therapy, extractions, and interceptive orthodontics are heavily dependent on a child's dental age rather than chronological age, this model could significantly aid general dentists and non-specialist providers in treatment planning and timing decisions. Future research should explore multicenter validation of YOLOv11 across different populations and age ranges, as well as integration into clinical workflows. Additionally, combining radiographic analysis with other biometric markers or EHR data may enhance the precision and utility of dental age models.

From a cost–benefit perspective, the proposed YOLOv11-based model offers several advantages over conventional manual methods of dental age estimation. First, in terms of operational efficiency, the model delivers automated predictions within seconds, reducing the need for time-intensive expert staging and interpretation of panoramic radiographs. This efficiency may lower resource expenditures and increase throughput in busy clinical or forensic settings. Second, the clinical impact is notable: automation has the potential to minimize diagnostic variability and reduce human error, thereby improving the consistency and quality of patient care. Third, from an implementation standpoint, the integration of an AI-based system into existing digital radiography workflows is likely to be cost-effective compared with repeated manual assessments by specialists, especially in resource-limited environments. While a full economic evaluation was beyond the scope of this study, these benefits suggest that the model could enhance accessibility and scalability of dental age estimation in clinical practice. Future work should incorporate formal cost–benefit analyses to quantify these advantages and provide additional evidence to guide adoption.

In conclusion, the findings show that YOLOv11-based DL model could accurately classify pediatric panoramic radiographs into discrete DA groups, restricted to the pediatric population aged from 3 to 14. The results confirmed that the proposed model achieved high classification performance, with a Top-1 accuracy of 92.6% on the validation set and strong generalizability on an independent test set. These findings demonstrate that YOLOv11 is an effective tool for automated DA estimation across 11 clinically relevant age categories.

## Supporting information

**S1 File. TRIPOD-AI Checklist.**
(DOCX)

## Acknowledgments

We would like to sincerely thank Dr. Tara Zarabian, pediatric dentsit for her valuable assistance in resolving the conflicts between the first and second reviewers during the dental age estimation process.

## Author contributions

**Conceptualization:** Mehdi Salehizeinabadi.

**Data curation:** Mehdi Salehizeinabadi, Sara Arastoo.

**Methodology:** Nazila Ameli.

**Project administration:** Camila Pacheco-Pereira.

**Resources:** Kasra Kouchehbaghi, Saghar Neghab.

**Supervision:** Camila Pacheco-Pereira.

**Writing – original draft:** Nazila Ameli, Kasra Kouchehbaghi.

**Writing – review & editing:** Mehdi Salehizeinabadi, Nazila Ameli, Saghar Neghab, Ida M Kornerup, Camila Pacheco-Pereira.

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
