## [Decision Letter · Decision Letter 0]

5 Aug 2025

Response to Reviewers
Revised Manuscript with Track Changes
Manuscript
**Journal Requirements:**

1. Please ensure that your Ethics Statement is available in its entirety at the beginning of your Methods section, under a subheading 'Ethics Statement'.

2. Please upload separate figure files in .tif or .eps format. Also, remove the figures from your manuscript file but keep the legends.

3. In the online submission form, you indicated that “ The data that support the findings of this study are available from the University of Alberta but restrictions apply to the availability of these data, which were used under license for the current study, and so are not publicly available. Data are however available at dentrsch@ualberta.ca upon reasonable request and with permission of University of Alberta.”.

3. Uploaded as supplementary information.

**Additional Editor Comments (if provided):**

1. Adding comparative analysis of different deep learning architectures as mentioned by the reviewer.

2. Adding the two key points regarding the Discussion and Study Limitations section mentioned by the reviewer.

3. Addressing the points mentioned for Results, Discussion and Recommendation section by the reviewer.

**Reviewers' Comments:**

**Comments to the Author**

1. Does this manuscript meet PLOS Digital Health’s publication criteria?

Reviewer #1: Yes

Reviewer #2: Yes

2. Has the statistical analysis been performed appropriately and rigorously?

Reviewer #1: Yes

Reviewer #2: Yes

3. Have the authors made all data underlying the findings in their manuscript fully available (please refer to the Data Availability Statement at the start of the manuscript PDF file)?

Reviewer #1: Yes

Reviewer #2: No

4. Is the manuscript presented in an intelligible fashion and written in standard English?

Reviewer #1: Yes

Reviewer #2: Yes

Reviewer #1: Thank you for sharing this article with me. I found it genuinely engaging. The study offers a thoughtful and well-organized approach to dental age estimation using deep learning, and the topic itself is clearly important for clinical practice in dentistry. Still, there are a few issues that the authors should address, along with some sections that would benefit from further clarification.

Introduction

It is recommended to address the challenges of dental age estimation based on current conventional methods in pediatric dentistry, along with their associated limitations and clinical sensitivities.

Methods section

Given that conventional machine learning methods are generally unsuitable for medical image processing, it is essential to clarify why simpler and more traditional machine learning algorithms were not employed despite the limited sample size, and why the authors opted for deep learning techniques after expanding the dataset, all while strictly adhering to scientific principles. Furthermore, the rationale for not comparing traditional machine learning methods with the implemented deep learning approach in this study must be explicitly explained. It is plausible that conventional machine learning models could achieve accuracy levels comparable to the proposed model, which should then be thoroughly analyzed and discussed in the Discussion section.

Including a comparative analysis of different deep learning architectures—like CNN, ResNet, and EfficientNet—would significantly strengthen the paper and clarify the findings for readers. Without such a comparison between various DL-based approaches, the current study leaves room for uncertainty about whether alternative methods might yield different results. Addressing this gap would enhance the paper’s overall value and credibility.

Dataset Augmentation and Preprocessing section shows:

The paper mentions how the images are distributed across different age groups. A table or a chart displaying the class distribution would be better than text to show the existing imbalance because it would visually demonstrate the problem. Visualizing the data allows readers to better grasp the problem of unbalanced data distribution.

The paper should address two key points regarding the Discussion and Study Limitations section in the following manner:

1. The negative effect of sample imbalance on model performance for age groups with limited data should be discussed.

2. The paper should explain whether it considered any specific strategies such as weighted loss functions or targeted data augmentation for minority classes to mitigate this issue.

In the second paragraph on page 6, data augmentation techniques are mentioned. It is recommended to provide more specific parameters for these techniques (e.g., the exact ranges for brightness/contrast variation) to:

1. Enhance readers' understanding of the preprocessing pipeline, and

2. Improve the reproducibility of the study's results.

Results

1. Line 174:

o The statement "A total of X panoramic radiographs was explored..." should either:

a) Replace X with the exact number of samples, or

b) Justify if the precise number cannot be disclosed.

2. Lines 188-189:

The observation that "a few misclassifications were observed between adjacent age groups…" is worth more consideration. A few ideas to include more discussion might be:

• Patterns: Are misclassifications concentrated in specific age groups?

• Anatomical Correlates: Were misclassifications typical linked to any specific anatomical features?

• Mitigation Strategies: What can be done to reduce these errors in future research?

• Clinical implications:

o If clinically insignificant, explain how this supports the methodological approach.

o If clinically significant, provide a fuller discussion of implications.

Discussion:

(Lines 246-247): The statement "our model achieved higher precision via regression analysis and automated full-image processing" seems to contradict:

1. The paper's claim of "direct classification" (Line 221), and

2. The actual output format of the model, i.e., 11 discrete age categories.

We recommend the authors to:

• Clarify this methodological discrepancy or explain if regression analysis was ever included in the pipeline

Lines 289-290:The phrase "restricted to the pediatric population aged from X to Y" calls for clarification.

Recommendation:

As a recommendation, one should consider carrying out cost-benefit analyses to create further clinical relevance for this study. In particular, the discussion should be framed in terms of the following points:

1. Operational Efficiency:

How the proposed model reduces time and resource expenditures for dental age estimation when compared to conventional methods

2. Clinical Impact:

Possible reduction of diagnostic errors

Increase in the quality of patient care

3. Implementation Value:

Cost-effectiveness of implementation of this AI in the clinical setting

Compare with manual assessment workstreams

Reviewer #2: The paper addresses an important topic with a novel application for pediatric dental age classification. However, there are certain areas where the manuscript can be improved utilizing the TRIPOD +AI checklist:

1. Dataset and Code availability: Both are not provided, which diminishes reproducibility of the research.

2. Outliers: There is no discussion about outliers. Although outliers are typically for numerical data, mentioning it would be helpful.

3. Calibration performance: The model’s performance should also be reported for calibration.

**Do you want your identity to be public for this peer review?** For information about this choice, including consent withdrawal, please see our Privacy Policy

Reviewer #1: No

Reviewer #2: **Yes: ** Shrey Lakhotia

**Figure resubmission:****Reproducibility:** To enhance the reproducibility of your results, we recommend that authors of applicable studies deposit laboratory protocols in protocols.io, where a protocol can be assigned its own identifier (DOI) such that it can be cited independently in the future. Additionally, PLOS ONE offers an option to publish peer-reviewed clinical study protocols. Read more information on sharing protocols at https://plos.org/protocols?utm_medium=editorial-email&utm_source=authorletters&utm_campaign=protocols

---

## [Editor Report · Decision Letter 1]

29 Sep 2025

Response to Reviewers
Revised Manuscript with Track Changes
Manuscript
**Journal Requirements:**
**Additional Editor Comments (if provided):**
**Reviewers' Comments:**
**Figure resubmission:**

**Reproducibility:** To enhance the reproducibility of your results, we recommend that authors of applicable studies deposit laboratory protocols in protocols.io, where a protocol can be assigned its own identifier (DOI) such that it can be cited independently in the future. Additionally, PLOS ONE offers an option to publish peer-reviewed clinical study protocols. Read more information on sharing protocols at https://plos.org/protocols?utm_medium=editorial-email&utm_source=authorletters&utm_campaign=protocols

---

## [Editor Report · Decision Letter 2]

18 Oct 2025

Dental age prediction from panoramic radiographs using machine learning techniques

PDIG-D-25-00375R2

Dear dentist Salehizeinabadi,

We are pleased to inform you that your manuscript 'Dental age prediction from panoramic radiographs using machine learning techniques' has been provisionally accepted for publication in PLOS Digital Health.

Best regards,

Shrey Lakhotia

Academic Editor

PLOS Digital Health